# Multiplexed imaging mass cytometry reveals distinct tumor-immune microenvironments linked to immunotherapy responses in melanoma

Xu Xiao [1,2,8], Qian Guo [3,8], Chuanliang Cui[3], Yating Lin [1], Lei Zhang[4], Xin Ding[5], Qiyuan Li[2,6], Minshu Wang[2,6], Wenxian Yang [7,9 ✉], Yan Kong [3,9 ✉] & Rongshan Yu [1,2,7,9 ✉]

**Abstract**

**Background** Single-cell technologies have enabled extensive analysis of complex immune composition, phenotype and interactions within tumor, which is crucial in understanding the mechanisms behind cancer progression and treatment resistance. Unfortunately, knowledge on cell phenotypes and their spatial interactions has only had limited impact on the pathological stratification of patients in the clinic so far. We explore the relationship between different tumor environments (TMEs) and response to immunotherapy by deciphering the composition and spatial relationships of different cell types.

**Methods** Here we used imaging mass cytometry to simultaneously quantify 35 proteins in a spatially resolved manner on tumor tissues from 26 melanoma patients receiving anti-programmed cell death-1 (anti-PD-1) therapy. Using unsupervised clustering, we profiled 662,266 single cells to identify lymphocytes, myeloid derived monocytes, stromal and tumor cells, and characterized TME of different melanomas.

**Results** Combined single-cell and spatial analysis reveals highly dynamic TMEs that are characterized with variable tumor and immune cell phenotypes and their spatial organizations in melanomas, and many of these multicellular features are associated with response to anti-PD-1 therapy. We further identify six distinct TME archetypes based on their multicellular compositions, and find that patients with different TME archetypes responded differently to anti-PD-1 therapy. Finally, we find that classifying patients based on the gene expression signature derived from TME archetypes predicts anti-PD-1 therapy response across multiple validation cohorts.

**Conclusions** Our results demonstrate the utility of multiplex proteomic imaging technologies in studying complex molecular events in a spatially resolved manner for the development of new strategies for patient stratification and treatment outcome prediction.

**Plain language summary**

Immunotherapies help the immune system to fight cancer. However, they only benefit a subset of melanoma patients, and currently no single marker is sufficient to determine which patients will respond to these treatments. Here, we use imaging mass cytometry, a technique to measure the levels of multiple markers in individual cells, to analyze tumor tissue from melanoma patients receiving immunotherapy. By determining the different cell types present and the spatial relationships between them, we identify six distinct melanoma cellular environments that are associated with different clinical responses to immunotherapy. Our results demonstrate how complex information about the spatial relationships of cell types can be integrated to help to identify patients that might benefit from immunotherapy.

[1] School of Informatics, Xiamen University, Xiamen, China. [2] National Institute for Data Science in Health and Medicine, Xiamen University, Xiamen, China. [3] Peking University Cancer Hospital and Institute, Beijing, China. [4] School of Life Science, Xiamen University, Xiamen, China. [5] Zhongshan Hospital, Xiamen University, Xiamen, China. [6] School of Medicine, Xiamen University, Xiamen, China. [7] Aginome Scientific, Xiamen, China. [8] These authors contributed equally: Xu Xiao, Qian Guo. [9] These authors jointly supervised this work: Wenxian Yang, Yan Kong, Rongshan Yu. ✉email: wx@aginome.com; k-yan08@163.com; rsyu@xmu.edu.cn

Advanced melanoma has a poor prognosis with a 5-year survival rate lower than 10%[1]. Immune checkpoint inhibitors (ICIs) targeting PD-1 and CTLA-4 have shown improved survival in advanced melanoma patients[1–4], but potent and durable response only presented in a subset of patients. To date, no single biomarker has been sufficient for patient stratification, presumably due to the complex immune response to cancer driven by both inter- and intrapatient cellular heterogeneities in tumor microenvironments (TMEs)[5,6]. Indeed, with deeper knowledge of the mechanisms of immune checkpoint blockade (ICB) based immunotherapy developed from recent clinical and preclinical studies, it is now recognized that ICI efficacy is driven by multifaceted interactions among a large diversity of cell lineages at both localized and systemic levels[7–15], thus defying the concept of patient stratification based solely on biomarkers that capture only limited dimensions of these intricate interactions.

In this study, we used imaging mass cytometry (IMC) to explore the composition and spatial arrangements of different immune and stromal cells in the vicinity of cancer cells in baseline tumor samples from 26 advanced melanoma patients treated with anti-PD-1 monoclonal antibody at Peking University Cancer Hospital and Institute (PUCH), Beijing, China. Using single-cell analysis on high-dimensional mass cytometry images, we quantified inter- and intra-tumor heterogeneities in a spatially resolved manner and identified important cellular features to classify melanoma into distinct archetypes linked to immunotherapy outcome.

## Methods

**Ethics statement**. The use of tumor samples in this study was approved by the Medical Ethics Committee of the Peking University Cancer Hospital and Institute (2019KT92). Written informed consent was obtained from each participant.

**Patient material**. A total of 55 formalin-fixed, paraffin-embedded (FFPE) tumor tissue samples were obtained from melanoma patients with anti-PD-1 monotherapy at Peking University Cancer Hospital and Institute (PUCH), Beijing, China. Patients were treated between March 2016 and March 2019, and their tissue samples were collected from untreated patients before anti-PD-1 monotherapy.

Twenty-nine tissue samples were excluded as they did not meet the IMC experimental requirements, yielding the final cohort of 26 samples in the study (14 responders and 12 nonresponders) (Supplementary Tables 1–3). Clinical data, including sex, age, overall survival (OS), progression-free survival (PFS), and clinical efficacy, were obtained from records of the patients with updated follow-up in Oct 2021 (Supplementary Table 2). PFS was defined as the time from the date of treatment to disease progression or last contact. OS was defined as the time from treatment to death or last contact. The clinical efficacy to anti-PD-1 monotherapy was evaluated by Response Evaluation Criteria in Solid Tumors (RECIST) version 1.1[16], including complete or partial response (CR/PR), stable disease (SD), and progressive disease (PD). All patients with CR/PR or SD were considered as responders and PD patients were considered as nonresponders.

**Antibody conjugation and validation**. An antibody panel of 35 proteins was designed to distinguish cell types and states, including immune, mesenchymal, proliferative, and immune checkpoint proteins (Supplementary Table 4). Twenty-five labeled antibodies were purchased from Fluidigm (https://www.fluidigm.com), and the remaining ten unlabeled antibodies were purchased from Abcam (https://www.abcam.com/). Antibodies

from Abcam were conjugated with metals using Maxpar X8 Multimetal Labeling Kit (Fluidigm, 201300) following the manufacturer's protocol. All conjugated antibody titration and specificity were tested by visual comparison of IMC images of some tissue slides from melanoma patients. Details about antibodies, metals, and concentration used in the study can be found in Supplementary Table 4.

**Preparation and staining**. Tissue slides were stained following IMC staining protocol (Fluidigm, PN400322) provided by Fluidigm. FFPE tumor samples were baked at 65 °C for 2 h to remove all visible wax. Slides were deparaffinized in fresh xylene (10 min twice) followed by rehydration through a graded alcohol series (100%, 95%, 80%, 70% for 5 min each). Antigen retrieval was conducted in a 96 °C water bath with Tris-EDTA buffer (pH 9.0) for 30 min. At room temperature (RT), slides were then blocked with 3% BSA in PBS (Maxpar) for 45 min in a hydration chamber after cooling to 70 °C. Meanwhile, the antibody cocktail was prepared in 0.5% BSA buffer mixed with the optimal dilution for each antibody (Supplementary Table 4). After blocking, slides were incubated with the antibody cocktail overnight at 4 °C in a hydration chamber. The next day, each slide was washed twice with 0.2% Triton X-100 in PBS (Maxpar), and twice with PBS (Maxpar). For DNA staining, slides were incubated with Intercalator-Ir (Fluidigm, 201192A) in PBS (Maxpar) at RT for 30 min. Finally, slides were washed with deionized water twice and air-dried at least 20 min before IMC acquisition.

**Imaging mass cytometry**. Images were acquired using a Hyperion Imaging System (Fluidigm). All operations were conducted following the manufacturer's procedure. Briefly, based on hematoxylin and eosin (HE)-stained serial tissue sections by a professional pathologist, we randomly selected regions of interest (ROIs) at the core tumor (CT) or invasive margin (IM) region (Supplementary Fig. 1; the number of ROIs per patient is listed in Supplementary Table 2). Images were laser ablated at 200 Hz, and raw data were acquired using a commercial acquisition software (Hyperion Imaging System, Fluidigm). The state of Hyperion Imaging System was monitored by the interspersed acquisition of data from the tuning slide (Fluidigm). We further asked our pathologist to determine whether these ROIs contain tertiary lymphoid structure (TLS), and 9 ROIs were determined as having TLS region based on its HE image and the protein (i.e., CD20, CD4, CD8, Ki67) expression pattern (Supplementary Fig. 2).

**IMC image processing, single-cell segmentation, and quantification**. We first checked the quality of every image by inspecting all marker staining patterns in the MCD Viewer (Fluidigm, v1.0.560.2). After quality control, a total of 158 images resulting in 662,266 single cells were used in the following analysis. Raw data (.mcd files) were converted to TIFF format using the imctools Python package (https://github.com/BodenmillerGroup/imctools). Then we used an in-house developed segmentation tool to perform single-cell segmentation on each image[17]. The mean expression of 35 proteins of the segmented single cells were extracted using the measure module in scikit-image (Python package, v0.16.2) by overlaying the generated segmentation masks on the corresponding TIFF images. To improve the accuracy of cell protein expression value, all images for each channel were processed by our developed quantification pipeline[18]. Briefly, for each protein channel, a large number of random decoy cells were generated from IMC image regions that likely contained noise only. We then subtracted the mean expression of the decoy cells from those of the segmented single cells to remove the effect of the background noise on the

quantification results. To remove the potential batch effect between ROIs, for each protein channel, we further identified positive cells by comparing the distribution of the expressions of the segmented single cells to that of the decoy cells with a false discovery rate (FDR) of 0.01, and normalized expressions of the segmented single cells across ROIs based on the expression of positive cells.

**Cell clustering analysis**. Single-cell protein expression data were clipped at the 99th percentile, followed by min–max normalization. For cell-type identification, 20 markers were used to define cell types: CD45, CD3, CD4, CD8a, FoxP3, CD20, CD68, CD14, CD16, CD11c, CD11b, IDO, Vimentin, $\alpha$-SMA, E-cadherin, EpCAM, CAIX, VEGF, PDGFRb, CollagenI. Three main cell types (lymphoid cells, myeloid cells, and other cells) were clustered and identified based on the protein expression pattern of each cluster. Then a second clustering was performed separately for each cell type on all markers except for the immune checkpoint proteins and PD-L1, resulting in 4 main cell types (lymphoid cells, myeloid cells, stromal cells, and tumor cells) and 20 distinguishable cell subtypes from 75 clusters. To obtain stable and robust cell clustering results[19], we followed the clustering pipeline from ref. [20] for our dataset (Supplementary Fig. 3a). Specifically, all clustering analyses were performed with two consecutive steps. First, meta-clusters were grouped with a self-organizing map implemented in FlowSOM[21] (R package, v1.18.0), and then Phenograph[22] (R package, v0.99.1) was applied on the mean expression values of each group from FlowSOM to obtain the final clustering results. Cell-type density was measured by the number of a certain cell type over total cells segmented from each image.

**Spatial analysis**. To investigate cell–cell interactions, a permutation-test method[23] implemented in neighbouRhood (R package, v0.3.0) was used to determine whether the interaction/avoidance between or within cell types occurred more frequently than random observation. Briefly, cells were classified based on their protein expression values by cell clustering analysis as mentioned above, then a null distribution of cell interaction pairs was generated with 1000 times permutation of random selection for each image. Observed cell interaction pairs were defined with a certain distance (20 μm between cell centroids). The $P$ value of interaction/avoidance between cell type A and B for each image was calculated as:

$$P_{AB} = \begin{cases} 1, & C_{obs} = 0; \\ \frac{\sum(C_{perm} \geq (\leq) C_{obs}) + 1}{N_{perm} + 1}, & \text{otherwise,} \end{cases} \quad (1)$$

where $C_{perm}$ is the number of cell pairs (A, B) in each permutation, $C_{obs}$ is the actual number of cell pairs (A, B) given a defined distance, and $N_{perm}$ is the number of permutation. $P$ values $\leq 0.01$ were considered as significant interaction/avoidance between cell types. Spatial proximity between two cell types were measured based on the distribution of the shortest distance from cells of one cell type to those of the other cell type on IMC images.

We further performed community analysis to identify common communities of multicellular units that existed across different TMEs[24]. Briefly, the IMC images were converted into topological neighborhood graph in which cells were represented as nodes and cell–cell neighboring pairs (20 μm between cell centroids) were represented as edges. Then we used the Louvain community detection method[25] to identify highly interconnected spatial subunits in the graph. This analysis was performed on all cells to uncover the microenvironment communities across samples. Phenograph (R package, v0.99.1) was then used to identify recurring similar spatial cell-type communities between samples based on minimum to maximum normalized percentages of cell types in each community.

**Measurement of intrapatient heterogeneity**. Each tumor sample represents a mixture of cells, including lymphoid, myeloid, stroma and tumor cells. We used Shannon entropy (H) to characterize intrapatient heterogeneity based on annotated cell subtypes from cell clustering results. To account for the different number of cells per sample, we subsampled 1000 cells from each sample $i$ for three times and calculated its Shannon entropy of each occurred cell-type frequency $P_c$ as:

$$H_i = -\sum_C P_c log_2(P_c). \quad (2)$$

This analysis was performed on samples with different regions to investigate the cell-type composition diversity in the CT or IM regions of patients using the Wilcoxon rank-sum test. We then compared the distribution of Shannon entropies of patients between responders and nonresponders.

**Identification of TME archetypes**. We first selected the cell types that were differentially enriched between responders and nonresponders ($log_2FC \geq 1.2$, adjusted $P \leq 0.05$), and with a cell-type density of at least 1% over total cells. The cell types that met these criteria were B, CD4$^+$ T, CD8$^+$ T, MC4, MC2, tumor (CAIX$^+$) cells for ROIs in the IM, and MC2 and MC4 cells for ROIs in the CT. Hierarchical clustering was then conducted separately for ROIs in the IM on the basis of the Euclidean distance on the selected cell-type abundances using hclust function with the Complete agglomeration method implemented in stats (R package, v3.6.3). For ROIs from the IM, six distinct groups were generated by cutree function (R package stats) with $k$ equal to 6. The resulting TME archetypes were further classified into two different categories (immune hot: H1, H2, and H3; immune cold: C1, C2, and C3) depending on their respective cell compositions. To characterize the TME for patients, we used majority voting on the basis of the TME archetype of their IM ROIs, and patients with equal numbers of cold and hot TME archetypes were considered as immune hot patients.

**Whole-transcriptome RNA sequencing and external public datasets**. The RNA-seq data of the PUCH cohort were obtained from our previous experiment[26], which were generated from unstained adjacent serial tissue slide from the same FFPE tumor samples used in this study for generating the IMC images. Sample RNA library construction and sequencing methods followed those as described in ref. [26]. Briefly, RNA-seq reads were mapped by STAR[27] and then quantified by RSEM[28] to get fragments per kilobase of transcript per million mapped reads (FPKM) values at the gene level. We further log2-transformed the read counts to avoid extremely skewed gene expression distributions.

In this study, we collected three RNA-seq datasets of melanoma patients treated with immunotherapy, together with their corresponding clinical information, including the Riaz17 ($n = 51$)[29], Gide19 ($n = 50$)[30], Liu19 ($n = 54$)[31] datasets (Supplementary Table 5). We used the immunotherapy outcomes provided in the original papers following RECIST guidelines. For the Gide19 and Liu19 studies, only samples that received anti-PD-1 monotherapy (nivolumab or pembrolizumab) were used. To obtain the gene expression data, we downloaded and processed the RNA-seq raw data by the same pipeline mentioned above for datasets Riaz17 and Gide19, and downloaded it from respective references provided by the authors for dataset Liu19.

**Identification of DEGs, pathway analysis, and prognostic score calculation**. Patients were classified into different TME archetypes based on majority voting, i.e., the archetype that had the most number of ROIs from a particular patient was assigned to the patient. Differential expression genes (DEGs) of each TME archetype were then identified using GLM function in edgeR (R package, v3.28.1) based on gene expressions of patients classified into that archetype vs. those of patients classified into archetypes of the opposite category. For example, DEGs of TME archetype H1 were derived based on gene expressions of patients from H1 vs. those from C1, C2, and C3. All DEGs with $\log_2FC \geq 1$ and $P \leq 0.05$ for each TME archetype were inputted into ClusterProfiler (R package, v3.14.3) for gene set enrichment analysis on hallmark gene sets in Molecular Signatures Database (MSigDB v7.4).

To derive a prognostic gene signature, we identified DEGs between immune hot and immune cold patients. By using the common genes between DEGs and Nanostring's IO 360 panel (770 curated cancer immune-related genes), we found 20 upregulated genes (*PLA1A, FAM30A, BLK, TDO2, CD19, MS4A1, GZMA, CCL19, FBP1, CD79A, TNFRSF17, CTLA4, CD7, CCL5, CDH1, CXCL9, CCL21, CD48, IL2RB, CD3G*) and 4 downregulated genes (*MAGEA4, FGF9, COL11A2, FZD9*). For each patient, the prognostic score was calculated as the ratio of mean expression of upregulated genes to that of downregulated genes.

**Deconvolution and ssGSEA score of 29 gene signatures from bulk RNA-seq data**. To estimate cell composition from bulk RNA-seq data, we used two deconvolution methods: MCP-counter[32] which uses average expression of canonical cell-type markers for cell-type abundance estimation, and CIBERSORTx[33], of which cell-type abundance is estimated using support vector regression on the basis of gene expression signatures of target cell types. We uploaded the normalized $\log_2$-transformed FPKM expression matrix on the MCP-counter website (http://134.157.229.105:3838/webMCP/) to get abundance scores for ten cell types. Immune cell frequencies of bulk RNA-seq data were inferred using CIBERSORTx (https://cibersortx.stanford.edu/) which uses gene expression profile matrices from scRNA-seq data for deconvolution. We uploaded PUCH RNA-seq data, selected the absolute mode with online provided melanoma scRNA-seq data as the signature matrix, disabled quantile normalization and applied 100 permutations for deconvolution robustness.

Single-sample gene set enrichment analysis (ssGSEA, Python implementation by Bagaev et al.[34]) was performed for 29 gene signatures which characterize four main TME groups (i.e., antitumor microenvironment, protumor microenvironment, angiogenesis fibrosis, and malignant cell properties)[34]. To account for local region bias of IMC data, the density of cell types for each sample were measured as the mean cell fraction of all ROIs taken from the same sample. Then we computed Spearman's rank correlation and R-squared of linear regression model between cell-type abundance from IMC and from RNA-seq data either by CIBERSORTx deconvolution or ssGSEA score of gene signature.

**Response prediction and survival analysis**. To validate the prediction performance for each dataset, Receiver Operating Characteristic (ROC) curve was drawn based on the prognostic score using sklearn (Python package, v0.22.2). Kaplan–Meier analysis was performed to estimate OS or PFS using survival (R package, v3.2.3). For each dataset, we separated samples into two groups based on their prognostic scores with thresholds determined automatically by survminer (R package, v0.4.7). The log-rank test was used to assess the statistical comparison between the

two groups, and a $P$ value $\leq 0.05$ was considered significant. Univariable Cox proportional-hazards models adjusted by age were used to estimate the prognostic factors on survival, and the hazard ratio (HR) of each factor was reported using survival (R package, v3.2.3).

**Statistics and reproducibility**. No statistical method was used to predetermine sample size, and sample selection of this study was based on sample availability. All analyses were conducted using software R (version 3.6.3) and Python (version 3.7). Association between response and melanoma subtypes was tested using Fisher's exact tests. The Wilcoxon rank-sum test was used for statistical analysis comparing continuous measurements, with Benjamini–Hochberg adjustment for all statistical tests involving multiple comparisons. An FDR-adjusted $P \leq 0.05$ was considered significant. All boxplots depict the median (the center line), interquartile range (IQR), and 1.5 times the IQR (whiskers), with outliers exceeding 1.5 times the IQR. For survival analysis, the statistical significance between Kaplan–Meier curves was tested by the log-rank test. Correlation between cell-type abundance was assessed by nonparametric Spearman's rank correlation. All statistical information used for experiments are defined in the figure legends.

**Reporting summary**. Further information on research design is available in the Nature Research Reporting Summary linked to this article.

## Results

**Global characteristics of cell compositions in melanoma TME**. To comprehensively characterize the microenvironment of melanoma patients with various stages and subtypes, we used a customized IMC panel of 35 antibodies on baseline tissue samples from 26 melanoma patients treated with anti-PD-1 (Fig. 1a and Supplementary Tables 1–3). In this cohort, melanoma subtypes were analyzed for association with immunotherapy outcome, and no factor shows a significant correlation with the clinical outcome, possibly due to our limited sample size (Supplementary Table 3). We then selected recognizing phenotypic markers of immune and stromal cell, immunoregulatory proteins, and proteins providing insights into cell activation and proliferation status (Supplementary Table 4). Regions of interest (ROIs) were randomly selected for each sample from core tumor (CT) and invasive margin (IM) regions based on hematoxylin and eosin (HE)-stained serial tissue section inspected by a professional pathologist. After quality control by manual inspections, 158 IMC images (59 from the CT: 34 responders, 25 nonresponders, 99 from the IM: 58 responders, 41 nonresponders) were further analyzed ("Methods" and Supplementary Fig. 1).

In total, 662,266 cells were clustered into 20 different cell subtypes using FlowSOM[21] and Phenograph[22] ("Methods"), which were further grouped into four major cell types, including lymphocytes, myeloid-derived monocytes, stromal cells, and tumor cells (Fig. 1 and Supplementary Fig. 3a, b). The lymphocytes included five different subtypes, namely, CD4$^+$ T cell (CD3$^+$CD4$^+$), CD8$^+$ T cell (CD3$^+$CD8$^+$), double-positive T cell (DPT; CD4$^+$CD8$^+$), T-regulatory cell (Treg; CD4$^+$FOXP3$^+$) and B cell (CD19$^+$) identified by their canonical cell markers. Myeloid-derived monocytes (MC1-MC6) were identified by CD14 and CD16, which can be classified into two categories based on their MHC Class II molecule (HLA-DR) expression. The first category included three subtypes characterized with highly elevated HLA-DR expression (MC4-MC6), indicative of their potential role as antigen-presenting cells (APC) within TME. Among them, subtype MC4 was further characterized with elevated dendritic cell marker CD11c and MC6

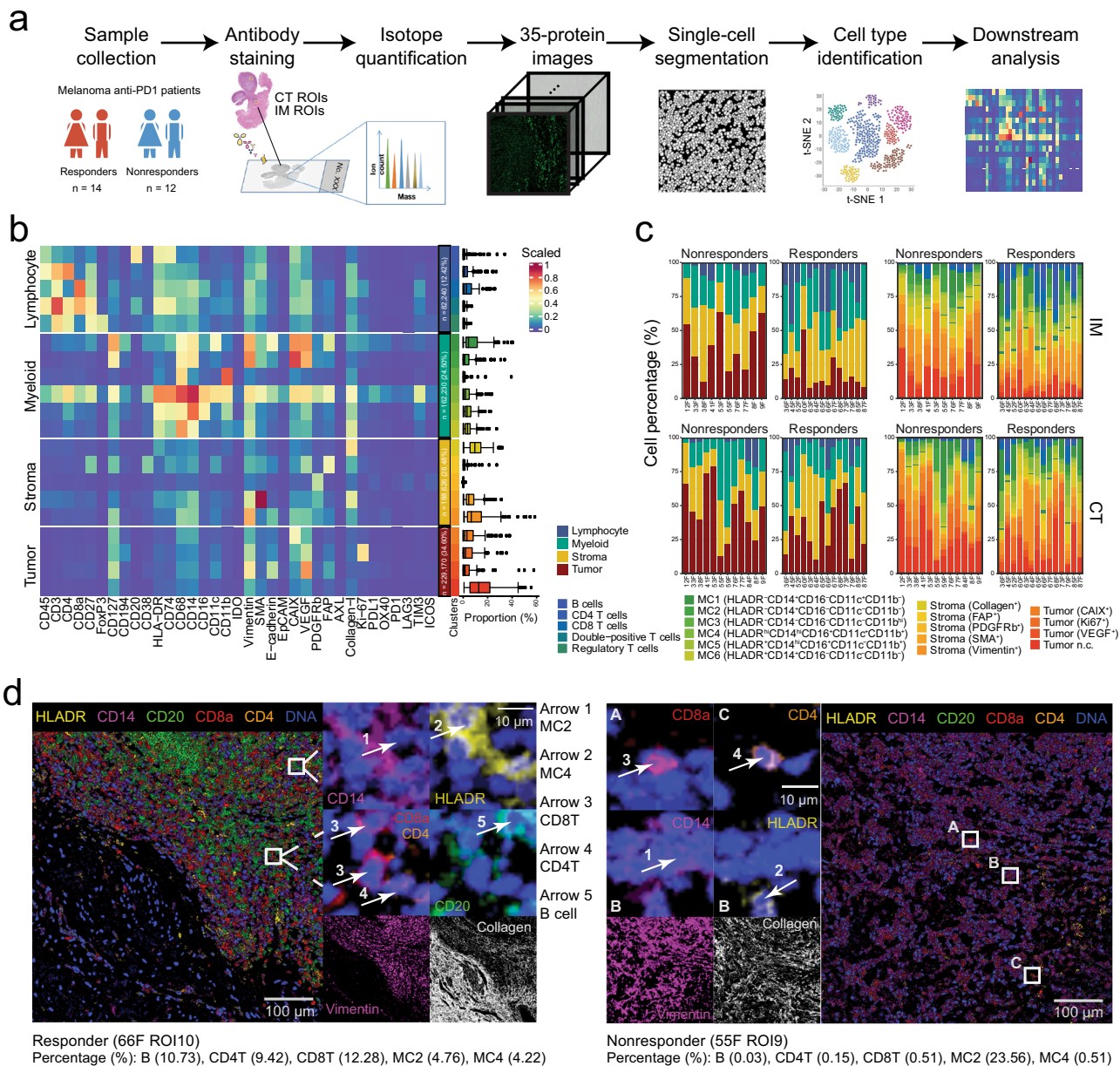

**Fig. 1 Overview of the study of melanoma patients using imaging mass cytometry (IMC) and characteristics of cell composition in tumor microenvironments. a** Workflow of IMC images acquisition from melanoma patients and data analyses. **b** Heatmap of mean values of scaled protein expression per cell type identified by unsupervised clustering (FlowSOM and Phenograph) for a total of 662,266 single cells. The boxplots on the right depicting the cell proportion of each IMC image. Each boxplot is shown with the median (the center line), interquartile range (IQR), and 1.5 times the IQR (whiskers), with outliers exceeding 1.5 times the IQR ($n = 158$ images). **c** Stack bars showing averaged cell percentage in images in the invasive margin (IM, top) and core tumor (CT, bottom) from responders and nonresponders, colored by four main cell types (left) and 20 cell subtypes (right). **d** Representative multichannel IMC images (ROI: region of interest) from one responder (left) and one nonresponder (right). Vimentin (magenta) and collagen I (white) were used to portray the structure of the tissue.

with elevated macrophage marker CD68. The second category was comprised of HLA-DR⁻ subtypes with elevated expression of exhaustion markers CAIX and VEGF (MC2, MC3) or indoleamine 2,3-dioxygenase 1 (IDO-1; MC1), representing their potential immune-suppressive roles as myeloid-derived suppressor cells (MDSCs). Stromal cells consisted of five subtypes denoted as S1 to S5 for Collagen⁺, FAP⁺, PDGFRb⁺, SMA⁺, and Vimentin⁺ cells, respectively, and tumor cells included 4 subtypes denoted as T1 to T4 for CAIX⁺, Ki67⁺, VEGF⁺, and a non-classified subtype (n.c.) that did not show the elevated expression on any markers from the defined panel, respectively.

All major cell types and subtypes were observed in all patients but with variation in cell compositions among patients and different tumor regions (Fig. 1c). Overall, the IM demonstrated more diversified cell-type compositions as indicated by higher Shannon entropy ("Methods") than the CT for both responders and nonresponders (Supplementary Fig. 3c). Furthermore, Shannon entropy analysis indicated more diversified cell-type compositions in responders than in nonresponders in the IM, but not in the CT. Two IMC images to exemplify TMEs with typical immune cells in a responder and a nonresponder are shown in Fig. 1d.

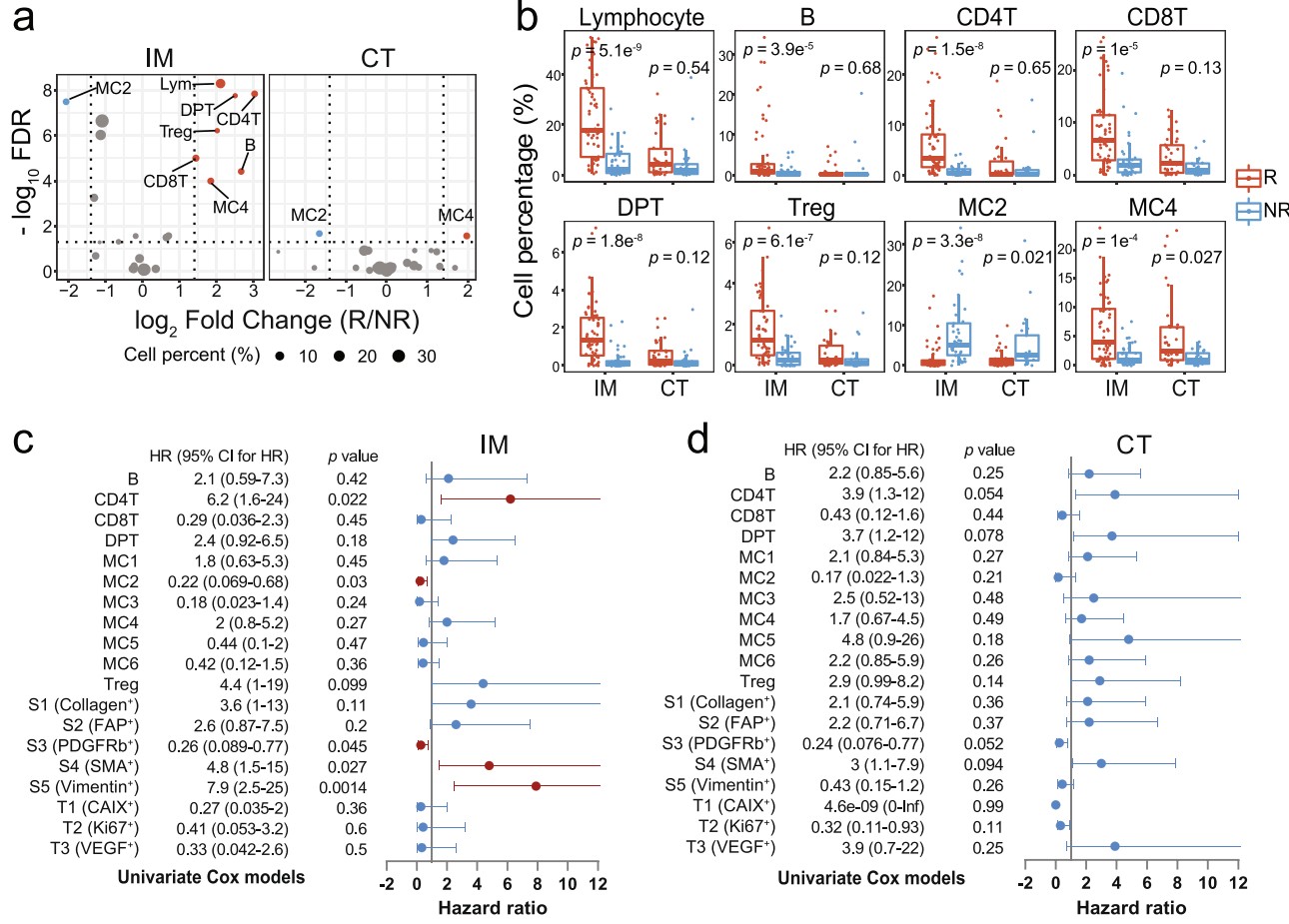

**Fig. 2 The prognostic impact of cell phenotypes density. a** Volcano plots showing differential testing of cell abundance in the invasive margin (IM, left) and core tumor (CT, right) between responders (R) and nonresponders (NR). The color of the nodes represents significantly higher abundance (red) and lower abundance (blue) of cell type in responders. The size of the nodes displays the percentage of cell type. **b** Boxplots showing the proportion of cell type in regions of interest from R (red) and NR (blue). Each boxplot is shown with the median (the center line), interquartile range (IQR), and 1.5 times the IQR (whiskers), with outliers exceeding 1.5 times the IQR. Points in the boxplot represent the cell percentage of each image. Comparisons were performed using Wilcoxon rank-sum test and adjusted with Benjamini–Hochberg method. **a**, **b** n = 58 for R group and n = 41 for NR group in the IM, n = 34 for R group and n = 25 for NR group in the CT. **c**, **d** Forest plots showing hazard ratios (nodes) and 95% confidence intervals (horizontal lines) of progression-free survival for each cell type in the IM (n = 24 patients) (**c**) and CT (n = 26 patients) (**d**) by univariate Cox models adjusted for age. The red nodes represent the significant factor with P value < 0.05. MC1: HLA-DR$^-$CD14$^+$CD11c$^+$ myeloid cells, MC2: HLA-DR$^-$CD14$^+$ myeloid cells, MC3: HLA-DR$^-$CD11b$^{hi}$ myeloid cells, MC4: HLA-DR$^{hi}$CD14$^{hi}$CD16$^+$CD11c$^+$CD11b$^+$ myeloid cells, MC5: HLA-DR$^+$CD14$^{hi}$CD16$^+$CD11b$^+$ myeloid cells, MC6: HLA-DR$^+$CD14$^+$ myeloid cells.

**Cell phenotype proportions differentiate TMEs of responders and nonresponders.** Examination of abundances of individual cell clusters from different TMEs revealed different cell compositions in TMEs from responders and nonresponders. The percentages of lymphocytes were significantly higher in responders than in nonresponders in the IM but not CT (Fig. 2a, b). A similar trend was observed in all five lymphocyte subtypes, indicating the important role of the IM in identifying TMEs that would respond to immunotherapy. Interestingly, despite the well-established immunosuppressive role of Treg, significantly elevated Treg densities were observed in the IM of responders compared to nonresponders, which were possibly recruited to the site for maintaining immunological unresponsiveness to self-antigens and suppressing excessive immune responses detrimental to the host. As a result, high abundance of Treg could indicate the presence of highly immunogenic tumor-associated antigens that would be able to induce a T-cell-mediated immune response after ICB for cancer rejection. For myeloid cells, we identified that HLA-DR$^+$ myeloid cells MC4 were significantly more abundant in responders, while HLA-DR$^-$ myeloid cells MC2 were

significantly enriched in nonresponders, and the difference can be observed in both the IM and CT. We also found that tumor cells with hypoxia signals (CAIX$^+$) were significantly enriched in the IM from nonresponders compared to responders, but this difference was not observed in the CT (Supplementary Fig. 4). No significant differences in other cell-type abundances were observed.

Cox regression analysis further revealed that the abundance of several cell types in the IM were associated with immunotherapy outcome. In the IM, CD4$^+$ T cells, SMA$^+$ stromal cells (S4), and Vimentin$^+$ stromal cells (S5) were associated with better outcome, whereas HLA-DR$^-$ myeloid cells (MC2) and PDGFRb$^+$ stromal cells (S3) were indicative of poor outcome after adjusted for age (Fig. 2c). None of the identified cell phenotypes in the CT was prognostic (Fig. 2d).

**Characteristics of immune checkpoint expressions in TME.** We next investigated the expressions of checkpoint molecules on different cell subtypes within TMEs to see if the compositions of any cell subtypes are associated with outcome to ICI treatments.

Overall, PD-L1 was expressed on a broad class of tumor and stromal cells within TMEs from both responders and non-responders, with MC4 having the highest average of PD-L1+ proportion (Supplementary Fig. 5a). However, none of their relative abundances, i.e., the percentages of PD-L1+ cells among the corresponding cell subtypes, was associated with response. Instead, significantly higher relative densities of PD-1+ CD4+ T and CD8+ T cells were observed in the IM of responders than that of nonresponders (Supplementary Fig. 5b), which is consistent with previous results that the fraction of exhausted cytotoxic T lymphocytes expressing high levels of CTLA-4 and PD-1 strongly correlates with response to anti-PD-1 in human melanoma[35].

In addition to PD-1, we also observed increased relative abundances of CD27+ and TIM-3+ cells among a broad class of lymphocyte and myeloid subtypes in the IM of responders (Supplementary Fig. 5c, d). CD27 is typically upregulated in the memory phenotypes of T cells upon exposure to stimulation[36]. In addition to their assumed roles in local immunity control, memory CD8+ T cells can further orchestrate the generation of systemic antitumor immunity by triggering antigen spreading through DC[37], and the presence of resident memory T cells is associated with durable response to immunotherapy in metastatic melanoma[38]. TIM-3 is a checkpoint receptor expressed on immune cells from TME including interferon (IFN)-γ-producing T cells and other leukocytes as well including DC and natural killer (NK) cells[39]. Although elevated expression of TIM-3 within TME was typically associated with T-cell exhaustion, a recent study showed that lack of TIM-3 expression of T cells may indicate a specific dysfunction status of T cells from ICB-refractory TMEs despite a brisk T-cell infiltrate[40]. In addition, a preclinical study using a murine model of head and neck cancer showed that the suppressive activity of TIM-3 can be reversed by IFN-γ secreted by CD8+ T cells upon PD-1 blockade[41]. These observations, together with the results described earlier, suggest the potential clinical utilization of predicting outcome to PD-1 based ICB therapy based on signatures of activated or previously activated antigen-experienced lymphocytes in the IM of tumor.

**Spatial analysis reveals heterogeneous cell–cell interactions in melanoma TME.** We performed regional correlation analysis to investigate the potential spatial co-occurrence patterns of different cells across all images. To avoid the potential nuisance effect of the absolute abundance of each cell type on the co-occurrence analysis, we used permutation-test-based neighborhood analysis[23] to identify statistically significant interaction or avoidance between pairs of cell types ("Methods," Fig. 3a, b; examples of cell–cell interactions and avoidance are shown in Fig. 3c–g and Supplementary Fig. 6a, respectively). Notably, subtypes of lymphocytes (CD4+ T, CD8+ T, DPT, Treg, and B cells) tended to form dense compartments with strong cognate interactions, and their proportions were highly correlated across images in responders (Fig. 3a, highlighted area I and Fig. 3c, d). In nonresponders, although the positive correlations between different lymphocyte subtypes were still maintained, co-locations of these lymphocytes, particularly between CD4+ T and other T-cell subtypes, were observed in fewer ROIs (Fig. 3b, highlighted area I and Supplementary Fig. 6b), indicative of a more diffused distribution of lymphocytes in these TMEs. We also observed highly different interaction patterns of HLA-DR+ and HLA-DR− myeloid cells with lymphocytes. Strong cognate interaction between the HLA-DR+CD11c+ myeloid cells (MC4) and lymphocytes can be observed in responders (Fig. 3a, highlighted area II and Fig. 3e) and, to a lesser extent, in nonresponders as well (Fig. 3b, highlighted area II and Supplementary Fig. 6b). In

contrast, significant interaction/avoidance between HLA-DR− myeloid cells and lymphocytes were observed in a much smaller number of ROIs (Fig. 3a, b, highlighted area III). Interestingly, in nonresponders, HLA-DR− myeloid cells showed avoidance to most lymphocytes except for MC1 and MC2, which showed interaction with CD8+ T cells (Fig. 3b, highlighted area IV). Significant proximate interaction between SMA+ stromal cells, which are primarily vascular smooth muscle cells that surround lymphatic vessels or blood vessels, and a broad class of immune cells were observed in most ROIs from both responders and nonresponders (Fig. 3a, b, highlighted area V and Fig. 3f, g), indicative of the important role of lymphovascular structures in maintaining the immune cell populations in TME.

**Different TME archetypes based on multicellular compositions.** We investigated how to translate the composition of single cells within TMEs into better stratification of melanoma to identify patients for immunotherapy. Using unsupervised hierarchical clustering on all the IMC images based the abundances of cell phenotypes that significantly differ in the IM regions of responders and nonresponders, we obtained six TME archetypes that demonstrated distinct cell compositions, including three immune hot TMEs characterized by strong infiltration of CD4+ T and B cells (H1), HLA-DR+CD11c+ myeloid-derived cells (H2), and CD8+ T cells (H3), respectively, and three immune cold TMEs with enrichment of CAIX+ tumor cells (C1), HLA-DR −CAIX+ myeloid-derived cells (C2), and an archetype with no significant enrichment of any cell type (C3), respectively (Fig. 4a; HE and IMC images of example ROIs from each TME archetype are shown in Fig. 4c). Signal pathway analysis with bulk RNA-seq data from paired samples also identified shared and distinct pathways of different TME archetypes (Fig. 4d and Supplementary Fig. 7). As expected, all immune hot TMEs showed multiple elevated signaling pathways that are correlated with adaptive and innate immune activation including IFN-α/γ response, allograft rejection, and complement pathway activities. H1 and H3 further showed an unregulated inflammatory response and KRAS upsignaling pathways, while H2 was uniquely enriched for hallmarks of p53 pathway, and H3 uniquely enriched for hallmarks of apoptosis, IL2-STAT5, and IL6-JAK-STAT3 pathways. Immune cold TMEs were predominantly enriched for signaling pathways typically associated with cancer progression or immune evasion, such as epithelial–mesenchymal transition and KRAS downsignaling.

We further performed community analysis[24] to investigate if single cells were organized differently in different TMEs. Using Louvain community detection[25] to identify communities of multicellular units that were physically contacted with each other, followed by unsupervised clustering based on their cellular compositions using Phenograph, we obtained 19 common communities across all images ("Methods" and Supplementary Fig. 8a). Close examination on the community composition of different TME archetypes showed that each archetype had its own predominant multicellular communities (Supplementary Fig. 8b). For the immune hot TMEs, H1 was dominated by Community 3 that constituted large networks of CD4+ T cells, B cells, and CD8+ T cells (Supplementary Fig. 9a); while in H2, the dominant community was Community 18 (Supplementary Fig. 9b) enriched for myeloid cells, primarily the HLA-DR+CD11c+ subtype MC4; and the majority community in H3 is Community 11 (Supplementary Fig. 9c) comprised of CD8+ T cells that interacted with HLA-DR+ myeloid cells MC5. These multicellular communities were seldom found in immune cold TMEs. Instead, cold TME C1 contained the highest percentage of Community 6 (Supplementary Fig. 9d) that was characterized by

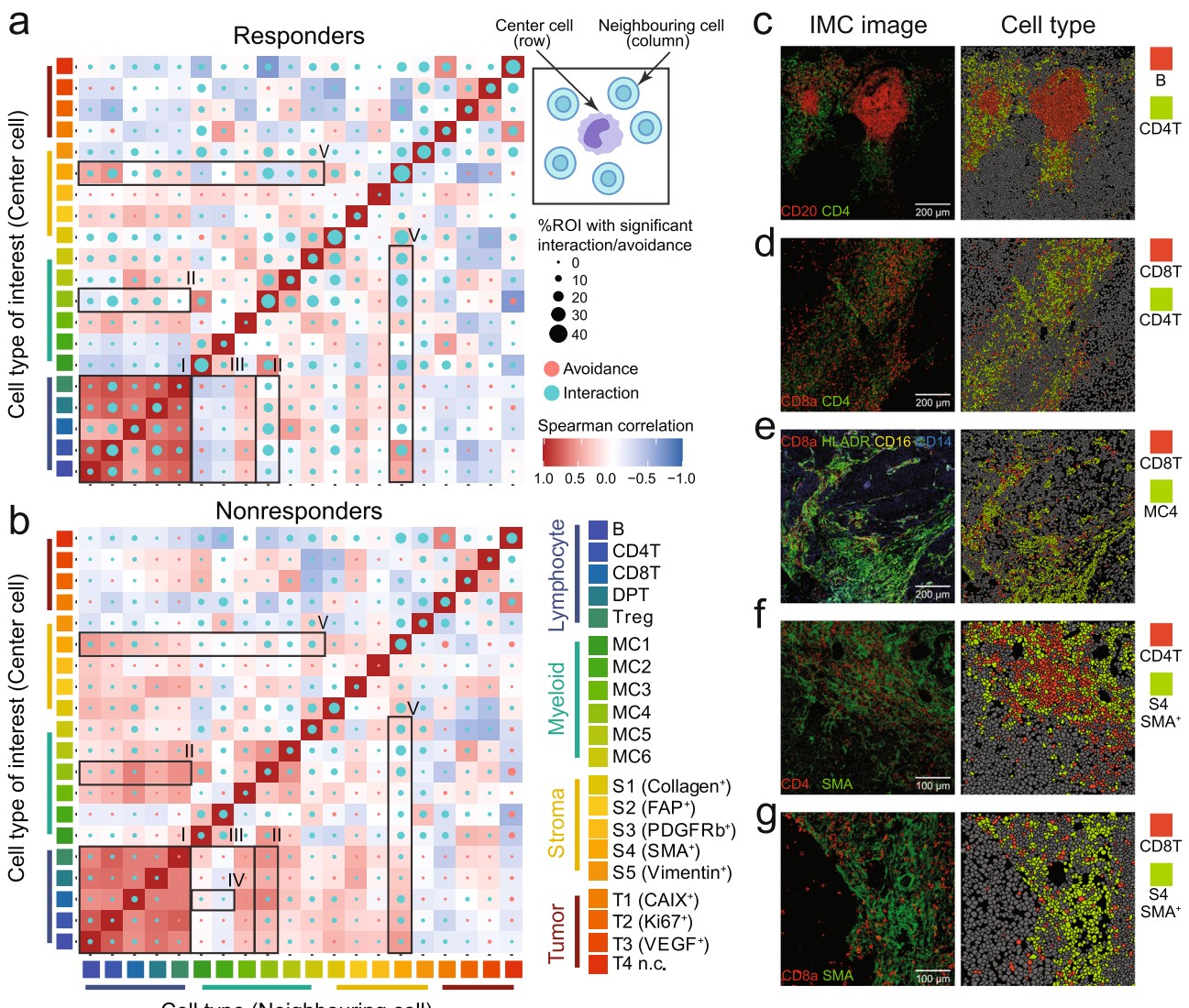

**Fig. 3 Spatial analysis among cell phenotypes. a, b** Circles indicating patterns of cell–cell interactions (green) or avoidances (red) for responders ($n = 99$ ROIs) (**a**) and nonresponders ($n = 59$ ROIs) (**b**). The circle size showing the percentage of images with significant interaction or avoidance determined by the permutation test ($P < 0.01$). Rows representing the relationship of all other cell types surrounding a cell type of interest. Columns representing the relationship of a cell type of interest surrounding other cell types. Color in heatmap squares indicating Spearman's rank correlation of cell types across all imaging mass cytometry (IMC) images in responders and nonresponders. Highlighted interactions or avoidance (numbered black boxes) include: (I) lymphocytes; (II) MC4 cells and lymphocytes; (III) HLA-DR⁻ myeloid cells and lymphocytes; (IV) MC1/MC2 cells and CD8⁺ T cells; (V) stromal SMA⁺ cells and immune cells. **c–g** Representative IMC images colored by marker (left columns) and cell type (right columns) showing the cell–cell interactions: **c** B cells are surrounded by CD4⁺ T cells, **d** CD8⁺ T cells are surrounded by CD4⁺ T cells, **e** CD8⁺ T cells are surrounded by MC4 cells, **f** CD4⁺ T cells are surrounded by SMA⁺ stromal cells, **g** CD8⁺ T cells are surrounded by SMA⁺ stromal cells. MC1: HLA-DR⁻CD14⁺CD11c⁺ myeloid cells, MC2: HLA-DR⁻CD14⁺ myeloid cells, MC3: HLA-DR⁻CD11bʰⁱ myeloid cells, MC4: HLA-DRʰⁱCD14⁺CD16⁺CD11c⁺CD11b⁺ myeloid cells, MC5: HLA-DR⁺CD14ʰⁱCD16⁺CD11b⁺ myeloid cells, MC6: HLA-DR⁺CD14⁺ myeloid cells. ROIs: regions of interest.

CAIX⁺ tumor cells in close contact with MC2 and Collagen⁺ stromal cells; and C2 was mostly dominated by Community 4 (Supplementary Fig. 9e) enriched for networks of Vimentin⁺ stromal cells and HLA-DR⁻VEGF⁺ myeloid cells MC2. Finally, cold TME C3 showed a highly diffused cell distribution without any dominant communities.

We asked if the above classification of TMEs was associated with clinical outcome to anti-PD-1. Overall, by dividing the clustering results into hot and cold categories, this clustering achieved a classification accuracy of 79.3% (46 out of 58 responder ROIs classified as immune hot) for responders and 95.1% (39 out of 41 nonresponder ROIs classified as immune cold) for nonresponders on the ROI level (Fig. 4b). Analysis

further revealed that ROIs from a same patient were in most cases highly homogeneous: most patients had ROIs from only one or two archetypes of the same immune hot or cold category (Fig. 4b). The exceptions included only two responders (79F and 63F) and one nonresponder (41F) who had ROIs from both immune hot and cold clusters. If we used majority voting to determine the TME archetype for each patient, all the 11 patients that were classified as immune hot were responders, representing an objective response rate (ORR) of 100%; and only 3 responders were from the immune cold patients, representing an ORR of 23.07%. Kaplan–Meier analysis revealed better overall survival (OS, $P = 0.0093$) and progression-free survival (PFS, $P = 0.06$) in patients defined as immune hot (Fig. 4e).

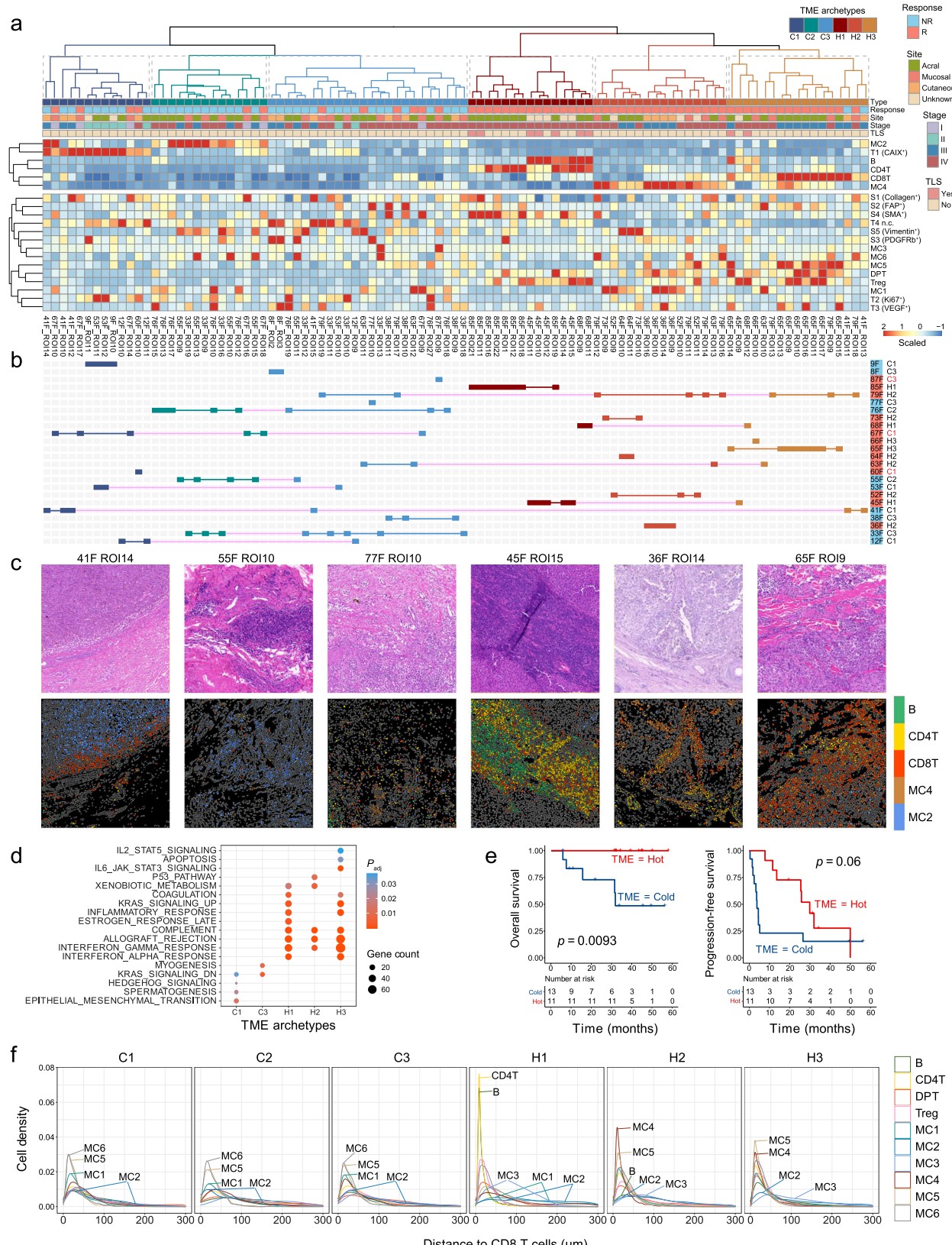

Interestingly, despite the recognized important role of CD8+ T infiltration to immunotherapy efficacy, only ROIs from H3 were characterized with high CD8+ T infiltration, representing only 6 out of 14 responders from this cohort. Close examination on different TMEs revealed highly different cell composition in the vicinity of CD8+ T cells (Fig. 4f and Supplementary Fig. 10). In immune hot TMEs, the dominant cells surrounding CD8+ T are either CD4+ T and B cells in H1 or HLA-DR+ myeloid cells MC4 in both H2 and H3, which is consistent with the recognized immune-enhancing actions governed by these cells. On the contrary, we observed elevated accumulation of the HLA-DR− subtypes of myeloid cells MC2 in close contact with CD8+ T cells

**Fig. 4 Identification of six distinct tumor microenvironment (TME) archetypes. a** Heatmap showing scaled cell-type abundance from the invasive margin regions of interest (ROIs). Six TME archetypes are clustered by the level of selected cell types (MC2, Tumor CAIX$^+$, B, CD4$^+$ T, CD8$^+$ T, and MC4 cells). **b** TME archetype patterns of each patient. If all ROIs from one patient are classified as having the same TME archetype, the patient is marked as the corresponding color of TME archetype. Patients who have ROIs that contain heterogeneous TME archetypes are indicated with magenta. **c** An example ROI from each TME archetype with hematoxylin and eosin (HE)-stained image (top) and its corresponding imaging mass cytometry image (bottom) with cell phenotyping (B, CD4$^+$ T, CD8$^+$ T, MC2, and MC4 cells). **d** Gene set enrichment analysis (GSEA) of genes upregulated expressed in patients with each TME archetype (the number of patients with C1, C2, C3, H1, H2, and H3 TME archetype are 6, 2, 5, 3, 6, and 2, respectively). Significantly enriched gene sets (adjusted $P < 0.05$, Benjamini–Hochberg method) from MSigDB HALLMARK collection are shown. There is no significant pathway enriched in samples with C2 TME archetype based on the 0.05 threshold for adjusted $P$ value. **e** Kaplan–Meier curves of overall survival (left) and progression-free survival (right) for melanoma patients based on their TME archetypes ($n = 13$ patients with immune cold TME, $n = 11$ patients with immune hot TME). $P$ values calculated using log-rank test. **f** Histograms showing the nearest distance in μm between CD8$^+$ T cells and other immune cells. MC1: HLA-DR$^-$CD14$^+$CD11c$^+$ myeloid cells, MC2: HLA-DR$^-$CD14$^+$ myeloid cells, MC3: HLA-DR$^-$CD11b$^{hi}$ myeloid cells, MC4: HLA-DR$^{hi}$CD14$^{hi}$CD16$^+$CD11c$^+$CD11b$^+$ myeloid cells, MC5: HLA-DR$^+$CD14$^{hi}$CD16$^+$CD11b$^+$ myeloid cells, MC6: HLA-DR$^+$CD14$^+$ myeloid cells.

in all three immune cold TME archetypes (Fig. 4f and Supplementary Fig. 10), indicative of the potential role of these cells in creating an ICI-resistant TME through suppressing effector T-cells functionality.

Recently, the tertiary lymphoid structure (TLS) formed in numerous tumor types is associated with improved clinical outcome[13,14,42,43]. To study if any of the above TME archetypes were associated with TLS, we asked a professional pathologist to identify TLS from the selected ROIs ("Methods"). In total, nine ROIs were determined as containing TLS (Supplementary Fig. 2). Among them, seven ROIs are from H1 TME (0.78, 95% CI: 0.40–0.97), two ROIs are from the H3 TME (0.22, 95% CI: 0.03–0.60), while none of these TLS was from immune cold ROIs (Fig. 4a). In addition, 46.7% of ROIs from H1 TME contain TLS. These results suggested that the immune hot TME archetypes identified in this study, in particular H1, were strongly associated with TLS.

**Gene signature derived from distinct TME archetypes predicts anti-PD-1 therapy response.** Recently, numerous gene expression signatures[33,34,44,45] have been developed to study the cellular composition of TMEs based on bulk RNA-seq data when single-cell information is not available. Here, we investigated the consistency between our single-cell analysis results from IMC data and the results from these signatures using RNA-seq data generated from adjacent serial sections from the same samples in the PUCH cohort[26]. We performed correlation analysis between 29 curated functional gene expression signatures (Fges)[34] and the cell-type abundances estimated by averaging over all IMC ROIs for each sample (Supplementary Fig. 11a). Interestingly, we found an over-representation of CD8$^+$ T-cells abundance in existing signatures despite that many of them have a putative target other than CD8$^+$ T cells. Among the 29 Fges, the Macrophage Fges shows the highest correlation with CD8$^+$ T-cells abundance in the paired sample, followed by Effector cell and T-cell Fges. Other than CD8$^+$ T cells, DPT abundance showed the strongest association with the Effector cell Fges, while Treg abundance showed the strongest association with the macrophage-associated Fges. Unfortunately, other than these three cell types, we did not find a strong association between the abundance of other cells and Fges. For example, no surrogate Fges for the abundances of myeloid subtypes were identified, while some myeloid cells (e.g., MC4), are strongly associated with clinical outcome to ICI in this study. Similar observations can be made when we compared the cell-type proportions estimated from IMC and those estimated by deconvolution methods from bulk RNA-seq data including MCP-counter[32], which derives cell-type abundance based on mean expression of canonical cell-type markers, and CIBERSORTx[33], of which cell-type abundance is estimated using support vector regression on the basis of gene expression signatures of target cell

types (Supplementary Figs. 11b, c). These findings suggested that existing RNA signature- and deconvolution-based methods for analyzing cellular compositions of TME could, at best, only capture the average cellular compositions of the whole tissue slide rather than their localized accumulations within TMEs due to the spatial heterogeneity of tumor tissues, while the latter are generally more essential for immunotherapy response prediction.

We further asked if it is possible to derive a global RNA-seq signature that could directly differentiate patients of different TMEs for immunotherapy outcome prediction without using cellular compositions as surrogates. To this end, we divided PUCH patients into immune hot and cold groups based on majority voting on their respective TMEs, and identified 20 significantly upregulated immune-related genes and 4 significantly downregulated immune-related genes in the immune hot group (Fig. 5a, b), where immune-related genes were defined as genes from the 770 curated cancer immune-related genes by Nanostring's IO 360 panel ("Methods"). We then calculated a response score as the ratio of mean expressions of 20 upregulated genes and 4 downregulated genes to measure the antitumor immunity level for predicting anti-PD-1 outcome.

To validate the performance of this signature, we analyzed RNA-seq data from the PUCH cohort[26] and three independent external datasets from melanoma patients treated with anti-PD-1 (Riaz17[29], Gide19[30], Liu19[31], Supplementary Table 5). The receiver operating characteristic (ROC) curve generated with clinical response data showed that the response score achieved an AUC of 0.83 (95% CI: 0.67–0.96) on PUCH, 0.75 (95% CI: 0.54–0.91) on Riaz17, 0.74 (95% CI: 0.59–0.89) on Gide19, and 0.65 (95% CI: 0.49–0.8) on Liu19, respectively (Fig. 5c). In addition, higher response scores were also associated with improved OS in the four datasets (Fig. 5d). Collectively, the above data demonstrated the potential value of using the response score derived from differentially expressed immune-related genes from patients of distinct TMEs as a biomarker for anti-PD-1 ICI treatments.

**Discussion**

Our multidimensional interrogation of baseline melanoma tissue samples before anti-PD-1 treatment provided a systematic landscape of immune microenvironments of melanoma patients with different response to immunotherapy. Importantly, our results revealed highly heterogeneous TMEs from responders to immunotherapy, and only a subset of these TMEs have high CD8$^+$ T-cells infiltration prior to immunotherapy, suggesting that anti-PD-1 therapy may have a much broader spectrum of mechanisms of action than only rejuvenating cytotoxic T cells that already reside in the TME. Indeed, rather than focusing on the specific state of a single-cell type, a comprehensive recognition on the contributions from all cell types relevant to effective anti-PD-1

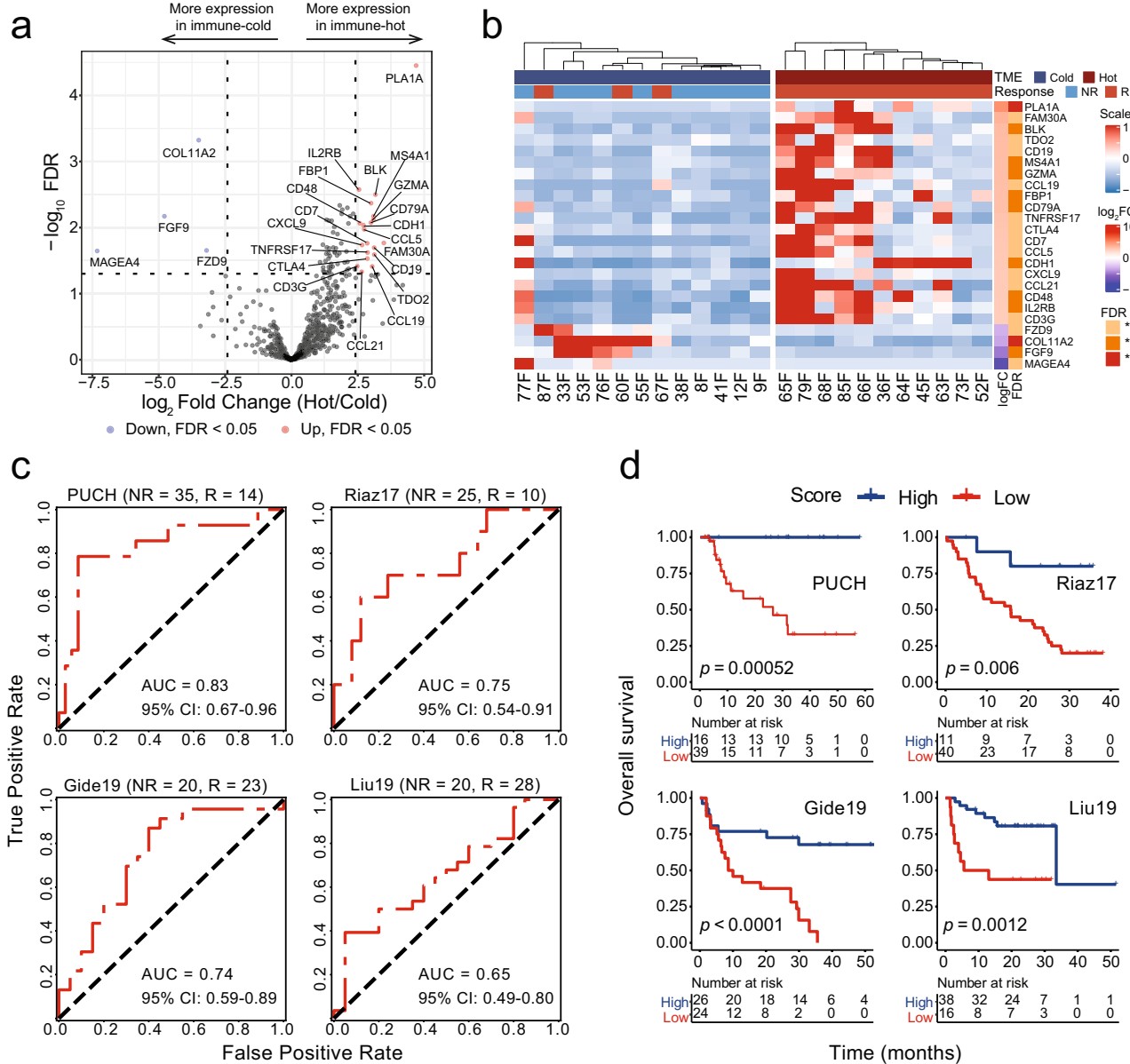

**Fig. 5 Prognostic impact of gene signature derived from tumor microenvironment (TME) archetypes. a** Volcano plot showing the upregulated genes (red) and downregulated genes (blue) in patients with immune hot TME. **b** Heatmap depicting the expression of 24 differential expression genes (DEGs, 20 upregulated genes, 4 downregulated genes) from PUCH patients classified as immune cold and immune hot groups. For panels (**a**) and (**b**), n = 13 for the immune cold group and n = 11 for the immune hot group. **c** The receiver operating characteristic (ROC) curve of sensitivity versus 1−specificity of the prediction performance of prognostic score calculated by the 24 DEGs for PUCH dataset (discovery cohort, nonresponsders NR = 35, responders R = 14)[26] and other three public datasets (i.e., Riaz17[29] (NR = 25, R = 10), Gide19[30] (NR = 20, R = 23), Liu19[31] (NR = 20, R = 28)). Patients with stable disease (SD) were not included. AUC area under curve. **d** Kaplan–Meier curves of overall survival in melanoma patients with high versus low prognostic score calculated by the 24 DEGs for PUCH dataset (n = 55)[26] and other three public datasets (i.e., Riaz17[29] (n = 51), Gide19[30] (n = 50), Liu19[31] (n = 54)). P values calculated using the log-rank test. PUCH Peking University Cancer Hospital and Institute.

activity would be required for developing successful biomarkers in immunotherapy. For example, it is now well recognized that helper CD4+ T cells play a pivotal role in generating effective immune responses[46,47] and CD4+ T-cell responses are required for optimal priming of antigen-restricted CD8+ T cells and their maturation[48]. Although PD-1 is thought to predominantly restrain CD8+ effector T cells, recent studies show that its' downstream effects further include activation of CD4+ T cells through targeting its costimulatory receptor CD28 by PD-1-recruited SHP2 phosphatase[49,50]. Moreover, recent studies demonstrate that pre-existing T cells in TME have limited reinvigoration capacity[51], and T-cell responses to ICB are mainly

derived from newly primed T-cell clones from extrinsic repositories such as tumor-draining lymph nodes (TDLN)[52], for which T-cell priming through APCs that acquire tumor antigen and migrate to the TDLN would be required[53,54]. For these reasons, as observed in the present study, enrichment of CD4+ T-cell and/or myeloid-derived APCs within TME could be a strong indicator to potential positive outcome to ICI in parallel to CD8+ T-cells infiltration.

Tumors have been previously classified into immune hot with strong immune cell infiltrates or cold with sparse infiltration, and these pre-existing immune states are related to their potential responses to immunotherapy[55]. Our results supported this

notion. Furthermore, empowered by multiplex single-cell image analysis, we were able to identify multiple immune archetypes from both immune hot and cold TMEs. Each archetype is made up with a unique cellular community composition and characterized by distinct dominant immune pathways, indicating that the previous TMEs delineations are incomplete to reveal the nuance of TMEs shaped by different tumor progression and immune evasion mechanisms. It is thus conceivable that such subdivision would enable us to further investigate the mechanisms behind different TME archetypes, from which better individualized therapeutic strategies based on archetypal assignments may be derived.

Although myeloid-derived cells are considered to associate with immune suppression within TME, it is now recognized that distinct myeloid cell subpopulations in the TME play different roles[56,57]. Consistent with this notion, our results revealed two highly distinct archetypes of TMEs enriched for different myeloid cells. The first archetype (H2), of which the TMEs were all from responders, showed enrichment of HLA-DR$^+$ myeloid cells, primarily the CD11c$^+$ subtype MC4, but low pre-treatment lymphocytes infiltration, suggesting a potential seminal role played by this group of myeloid cells in mediating an inflammatory microenvironment towards positive outcome from anti-PD-1 treatment. The second archetype (C2), which was associated with poor clinical outcome to ICI, showed elevated accumulation of subtypes of highly exhausted myeloid cells with low HLA-DR expression and elevated VEGF and CAIX expressions (MC2), confirming their roles in immune suppression. Hence, in developing combination therapy that targets both T-cell rejuvenation and macrophage depletion[58], e.g., through CSF1R inhibitors[59] combining with ICB therapy, it may be necessary to identify the right target patients based on the composition of their myeloid infiltration as these inhibitors may lack the specificity to differentiate between protumor and antitumor myeloid cell subsets. In addition, repolarizing myeloid cells within TME to sustain or restore their tumoricidal activities through engaging pathogen recognition receptors (PRRs)[60] or agonistic anti-CD40 antibody[61] could be a promising combination therapeutic strategy to improve clinical response to ICI treatments for patients with this archetype of cold TME.

Other than TMEs enriched with exhausted myeloid cells, our results indicated the existence of another distinct immune cold TME archetype derived primarily from nonresponders (C1). TMEs of this archetype did not show strong infiltration of myeloid cells but were characterized with enrichment of tumor cells with high expression of hypoxia signaling molecule CAIX. The hypoxic condition of tumor regions is typically arisen from increased oxygen consumption by rapidly proliferating tumor cells in combination with inadequate oxygen supply due to abnormal tumor angiogenesis[62]. Hypoxia-driven mechanisms allow tumor cells to continue to survive and proliferate in the hypoxic TME, while creating an inhospitable environment for immune cells through promoting apoptosis of T lymphocytes[63] and DCs[64], preventing effector T-cells activation[65] and their homing to the TME[63], and promoting immune-suppressive stromal cells differentiation[66], leading to tumor resistance to immunotherapy. Therefore, hypoxia may be exploited as a potential biomarker to identify this type of nonresponders, for whom strategies that combine methods to overcome hypoxia in cancer, including hypxia-activated prodrugs (HAPs)[67], inhibition of HIF signaling or its downstream pathways[68], or supplemental oxygenation[66,69] with immunotherapy may be explored.

The limitations of this study include the small size of the cohort and retrospective design. Nevertheless, our analysis has revealed highly heterogeneous multicellular features and their spatial interaction within a histological context of tumor TME,

and confirmed that many of these features are associated with the clinical benefit of immunotherapy. Our results thus provide the basis for future studies on multicellular structures based on spatially resolved single-cell data for an in-depth characterization of the tumor microenvironment, from which better methods to identify the right patients for different immunotherapy strategies can be derived. Moreover, our results further indicate that such knowledge is highly translatable, and can be exploited in multiple applications ranging from guiding the design of traditional bulk molecular tests for better patient segregation results despite their limitations in both spatial and single-cell resolutions, or identification of targets for the development of novel therapies.

## Data availability

Raw IMC images and processed data, including source data for the figures, are deposited in Zenodo with the identifier https://doi.org/10.5281/zenodo.6838169[70]. Previously published melanoma RNA-seq datasets reanalyzed here are referenced to and available accordingly (PUCH ($n = 55$)[26], Riaz17 ($n = 51$)[29], Gide19 ($n = 50$)[30], Liu19 ($n = 54$)[31]). All other data are available from the corresponding author on reasonable request.

## Code availability

Analysis codes required to reproduce the results are available in Zenodo with the identifier https://doi.org/10.5281/zenodo.6838169[70].

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

## Acknowledgements

This work was supported by the Natural Science Foundation of China (81972557), National Key R&D Program of China (2019YFA0904400), Beijing Natural Science Foundation (7202022), Beijing Municipal Science and Technology Commission (Z191100006619006), and the Fundamental Research Funds for the Chinese Central Universities (20720190101 to Q. Li).

## Author contributions

W.Y., Y.K., and R.Y. supervised the research and designed the methodology. X.X., Q.G., C.C., Y.L., and M.W. performed image quantification, analyzed data and generated figures. L.Z and X.X. performed all IMC experiments. X.D. provided confirmatory pathology analyses. Y.K., Q.G., and C.C. provided patient samples and clinical input of the study. The manuscript draft was done by X.X. and revised by W.Y., R.Y., Y.K., and Q.L. All authors have further assisted in reviewing the manuscript.

## Competing interests

W.Y. and R.Y. are shareholders of Aginome Scientific. The remaining authors declare no competing interests.
