## [Peer Review File · Communications Medicine]

Reviewers' comments:

Reviewer #1 (Remarks to the Author):

In this manuscript Xiao et al., have examined the immune landscape of baseline tumour biopsy specimen from melanoma patients in order to identify the immunological markers of response to anti-PD-1 treatment. The authors have identified some archetypes associated with response. Although findings are largely confirmatory to previously published work, the use of IMC has enabled the authors to do some in-depth analysis of potential cellular interactions that could contribute to the success of anti-PD1 therapies. Therefore, I believe this could be of value to the readership of this journal

However, there are some concerns that need to be addressed by the authors before consideration for publication

Major concerns

1. The cohort - It appears the authors have used a mixed bag of clinical samples, ranging from different stages of melanoma to different site of disease. Some of the differences can impact clinical outcomes and the authors need to clarify how they have addressed that. (a) Differences in type of melanoma - immunotherapy success rate is different between cutaneous melanoma and acral/mucosal melanoma, (b) different stages of disease - could the authors also confirm there were 4 patients in stage I & II who were treated with anti-PD-1?
2. Selection of ROIs - Can the authors clarify how the regions of interest were selected and how many ROIs per patient were analysed? Importantly, how representative these ROIs were of the whole tissue? This is also relevant for the selection of archetypes. How were the archetypes assigned to patients (based on how many ROIs)?
3. I note that no tumour cell markers were used in the IMC (for example SOX10). How did the authors define tumour cells/regions?
4. It will be useful if the authors explain the rationale for using first FlowSOM and then Phenograph for clustering, when both are used for the same?
5. It appears in Fig 1. the majority of the myeloid populations identified did not express CD45. Could the authors clarify?
6. Data on cellular interactions are interesting. However, cell numbers could play a part in this and could the authors clarify how this was avoided? For example, if 'hot tumours' had more lymphocytes, the chances of interactions between two lymphocytes are higher because of the numbers. Did the authors downsample (compare equal number of cells) like what was done for the measurement of inter-patient tumour heterogeneity?

Minor comments

1. As with any imaging techniques, identifying co-localisation vs potential image artifact is critical. Therefore when gating for populations, negative gating could be used to avoid some misrepresentations. I note that the authors have not done this, but could the authors explain how they avoided this issue? For example, on Fig 1, some myeloid cells appear to be expressing CD3. Is it

real or is it that a T cell is always adjacent to these myeloid cells?

2. Methods section on 'preparation and staining' is riddled with errors. Second line reads 65% instead of 65 degrees, line 6 states 70 degrees as room temperature.

3. It appears Figure 4D and 4E are mixed up in the figure legend

4. Figure 4D (in figure) doesn't have C2 (in Fig 4D).

Reviewer #2 (Remarks to the Author):

Xiao et al reported a imaging mass cytometry analysis, coupled with bulk RNA-seq, to analyze the spatial pattern of pre-selected immune, stromal and tumor populations in a cohort of 26 melanoma patients from PUCH. The authors discovered different tumor microenvironment patterns that are related to response and non-response to anti-PD-1 therapy. In particular, they reported three separate "hot" TMEs that enrich different immune cell types: a CD8 related, high HLA-DR related (APC) and CD4 T and B cell related. In the cold TME, the enriched biological process is hypoxia (in either tumor or myeloid subset MC1/MC2). Finally, based on tumor matched RNA-seq data, the authors came up with a gene expression signature that were shown to be predictive of survival in multiple immunotherapy datasets of melanoma.

The reviewer is convinced that this work is of high interest to the field and highlights the importance of spatial information to predict response to anti-PD-1, even when only using a pre-selected markers of 35 proteins. My major concern is that most of the raw data and supporting processed values are nowhere to be found, which forces the reviewer to accept the results in the figures as is. The uploaded data to the figshare also requires log in, which may compromise the anonymity of the review process.

Major comments:

1) The distribution of the number of CT and IM ROIs per patient should be provided.

2) Both the overall and progression free survival data for each patient should be provided, along with other clinical parameters like clinical response, disease stage, previous treatment etc, in a supplementary table.

3) The processed IMC data (cells after segmentation with raw marker intensity across 35 markers for all ROIs) should be provided to allow independent reproduction of the results listed here.

4) In the same vein, the reviewer cannot find the RNAseq data of this cohort being made available; the data should be included to ensure the reproducibility of the analysis. The github page only contain two R scripts. If the RNAseq is a part of the previous Cui et al publication (n=49 tumors), then the patient mapping from this set to the Cui et al set should be provided.

5) The proximity to CD8 T cell analysis (Fig 4f) is not easy to decipher - there is no way for the reviewer to see, in the current data presentation, that MC1 and MC2 are closer to CD8 T cells in the immune cold TMEs. Also, the MC1 and MC2 was discussed to be avoiding the lymphocytes in the discussion in Fig 3.

6) When the authors mentioned that the expression signatures of Bagaev et al or CIBERSORTx being uncorrelated with the IMC result, did the author tested basic correlation between the canonical T cell and myeloid specific transcript like CD8A/CD8B/CD14/CD3E etc between the bulk RNAseq and the IMC cell percentage?

7) Is the H1 TME correlated with tertiary lymphoid structure?

8) Could MC4 monocyte derived DCs? the cluster's HLA-DR expression is significantly higher than other MC clusters. Independent staining of serial section using CD14, CD1C, CLEC10A and HLA-DR.

9) The prognostic significance of the 24-gene should be compared to IFN- γ /IMS ratio that was reported in Cui et al.

Response to Reviewers' Comments

Dear Editor,

We are grateful to the editors and reviewers for their time and comments on our manuscript. Below, we provide a point-by-point response to editorial and reviewers' comments. In addition, the changes that we made to the original manuscript were highlighted in red in the revised manuscript.

Reviewer #1 (Remarks to the Author):

In this manuscript Xiao et al., have examined the immune landscape of baseline tumour biopsy specimen from melanoma patients in order to identify the immunological markers of response to anti-PD-1 treatment. The authors have identified some archetypes associated with response. Although findings are largely confirmatory to previously published work, the use of IMC has enabled the authors to do some in-depth analysis of potential cellular interactions that could contribute to the success of anti-PD1 therapies. Therefore, I believe this could be of value to the readership of this journal.

However, there are some concerns that need to be addressed by the authors before consideration for publication

Major concerns

1. The cohort - It appears the authors have used a mixed bag of clinical samples, ranging from different stages of melanoma to different site of disease. Some of the differences can impact clinical outcomes and the authors need to clarify how they have addressed that. (a) Differences in type of melanoma - immunotherapy success rate is different between cutaneous melanoma and acral/mucosal melanoma, (b) different stages of disease - could the authors also confirm there were 4 patients in stage I & II who were treated with anti-PD-1?

Author's response: In the revised manuscript, we have added the immunotherapy success rate among melanoma patients with different site and stages (Supplementary Table 3, see below). We further performed Fisher's exact tests to assess the association between response and types/stages of melanoma, which showed that there is no significant difference in response rate with different types/stages in this cohort. The results were summarized in the revised manuscript as follows:

“To comprehensively characterize the microenvironment of melanoma patients with various stages and subtypes, we used a customized IMC panel of 35 antibodies on baseline tissue samples from 26 melanoma patients treated with anti-PD-1 (Fig. 1a; Supplementary Tables 1-3). We first investigated the effect of tumor site and stage on immunotherapy outcome, and the results showed that there is no significant association between clinical outcome and melanoma type in this cohort (Supplementary Table 3).”

Table 3: Pathological response for all patients with different tumor site and stage.

	Overall, N (%)	Responder, N (%; 95% CI) ¹	Nonresponder, N (%; 95% CI) ¹	P value ²
Tumor site				0.94
Acral	11 (42.3%)	6, (54.5%, 23-83)	5, (45.5%, 17-77)	
Mucosal	6 (23.1%)	4, (66.7%, 22-96)	2, (33.3%, 4-78)	
Cutaneous	5 (19.2%)	2, (40%, 5-85)	3, (60%, 15-95)	
Unknown	4 (15.4%)	2, (50%, 7-93)	2, (50%, 7-93)	
Tumor stage				0.16
I	1 (3.8%)	1 (100%, 5-100)	0 (0, 0-95)	
II	3 (11.5%)	0 (0, 0-63)	3 (100%, 37-100)	
III	6 (23.1%)	3 (50%, 12-88)	3 (50%, 12-88)	
IV	16 (61.5%)	10 (62.5%, 35-85)	6 (37.5%)	

¹ 95% CI are measured by two-sided Clopper-Pearson exact method

² P values are based on Fisher's exact tests to test association between response and tumor type.

We also included the clinical information in Fig. 4a in the revised manuscript (as shown in below).

We checked the clinical data of 4 patients in stage I and II and confirmed that these patients were indeed treated with anti-PD1. Note that the stages of these patients were determined at the time of diagnosis and all patients in this cohort met the requirements for anti-PD1 treatment.

2. Selection of ROIs - Can the authors clarify how the regions of interest were selected and how many ROIs per patient were analyzed? Importantly, how representative these ROIs were of the whole tissue? This is also relevant for the selection of archetypes. How were the archetypes assigned to patients (based on how many ROIs)?

Author's response: It is a good question. In selecting the ROIs, we tried to ensure that the ROIs evenly distributed over IM and CT regions identified by a professional pathologist. In the revised manuscript, we added a new figure (Supplementary Fig. S11) to show the selected ROIs of all slides, and provided the number of CT and IM ROIs per patient in Supplementary Table 2.

To determine the patient level classification of TME archetype, we used majority voting based on TME archetype of their IM ROIs. This approach has now been explained in more details in the revised manuscript as follows:

“To characterize the TME for patients, we used majority voting on the basis of the TME archetype of their IM ROIs, and patients with equal numbers of cold and hot TME archetypes were considered as immune-hot patients.”

3. *I note that no tumour cell markers were used in the IMC (for example SOX10). How did the authors define tumour cells/regions?*

Author’s response: This is a very good question. We didn’t include tumor cell markers in our initial panel design, which is a limitation in our study. However, in selecting the ROIs, we have asked our pathologist to exam our ROI selection on the HE-stained serial tissue sections to make sure that all ROIs were from tumor regions.

4. *It will be useful if the authors explain the rationale for using first FlowSOM and then Phenograph for clustering, when both are used for the same?*

Author’s response: That is a good question. Both of them are unsupervised clustering methods. However, according to *Liu et al.* ^[1], PhenoGraph is more robust when detecting refined sub-clusters, whereas FlowSOM tends to group similar clusters into meta-clusters. Moreover, the performance of PhenoGraph is affected by large sample sizes, but FlowSOM is relatively stable as sample size increases. Therefore, following the approach in *Ali et al.* ^[2], we used FlowSOM to pre-cluster the original cells into a sufficiently large number of clusters, and then performed PhenoGraph on these clusters to obtain the final clustering results. In the revised manuscript, we added the following paragraph to explain the selection of the clustering approach used in this study:

“To obtain stable and robust cell clustering results ^[1], we followed the clustering pipeline from *Ali et al.* ^[2] (Supplementary Fig. S1a). Specifically, all clustering analyses were performed with two consecutive steps. First, meta-clusters were grouped with a self-organizing map implemented in FlowSOM (R package, v1.18.0), and then Phenograph (R package, v0.99.1) was applied on the mean expression values of each group from FlowSOM to obtain the final clustering results.”

[1] Liu, X. et al. A comparison framework and guideline of clustering methods for mass cytometry data. *Genome Biology* 20, 1-18 (2019)

[2] Ali, H. R. et al. Imaging mass cytometry and multiplatform genomics define the phenogenomic landscape of breast cancer. *Nature Cancer* 1, 163–175 (2020)

5. *It appears in Fig 1. the majority of the myeloid populations identified did not express CD45. Could the authors clarify?*

Author’s response: It is a good question. Indeed, in our result, the majority cells from clusters annotated as myeloid subtypes did not express CD45 (see Figure 1 below). However, most of these clusters were highly distinguishable from other clusters in terms of myeloid cells related markers (see figures below, CD14: Figure2, CD68: Figure 3, CD11b: Figure 4, CD11c: Figure 5). Based on this observation, we annotated these clusters as myeloid cells.

Figure 1. Normalized CD45 expressions in cell types

Figure 2. Normalized CD14 expressions in cell types

Figure 3. Normalized CD68 expressions in cell types

Figure 4. Normalized CD11b expressions in cell types

Figure 5. Normalized CD11c expressions in cell types

6. Data on cellular interactions are interesting. However, cell numbers could play a part in this and could the authors clarify how this was avoided? For example, if 'hot tumours' had more lymphocytes, the chances of interactions between two lymphocytes are higher because of the numbers. Did the authors downsample (compare equal number of cells) like what was done for the measurement of inter-patient tumour heterogeneity?

Author's response: It is a good question. The absolute abundance of a particular cell type could be a confounded factor for spatial analysis. Therefore, in our analysis we used the permutation-test approach suggested in Schapiro *et al* ^[1] to avoid this effect, which performs statistical methods to determine whether the interactions/avoidances between or within cell types occurred more frequently than random observation. Briefly, a null distribution of cell interaction pairs was generated with N (1000 in paper) times permutation of random selection for each image. Then the interaction/avoidance between cell types was compared to the null distribution. This is explained in the revised manuscript in more details as follows:

“We performed regional correlation analysis to investigate the potential spatial co-occurrence patterns of different cells across all images. To avoid the potential nuisance effect of the absolute abundance of each cell type on co-occurrence analysis, we used permutation-test-based neighborhood analysis ^[1] to identify statistically significant interaction or avoidance between pairs of cell types (Methods, Fig. 3a-b; examples of cell-cell interactions shown in Fig. 3c-g).”

[1] Schapiro, D. *et al.* histoCAT: analysis of cell phenotypes and interactions in multiplex image cytometry data. *Nature Methods* 14, 873 (2017).

Minor comments

1. As with any imaging techniques, identifying co-localisation vs potential image artifact is critical. Therefore when gating for populations, negative gating could be used to avoid some

misrepresentations. I note that the authors have not done this, but could the authors explain how they avoided this issue? For example, on Fig 1, some myeloid cells appear to be expressing CD3. Is it real or is it that a T cell is always adjacent to these myeloid cells?

Author's response: It is indeed a very good question. The IMC images can be very noisy if not properly processed. In our processing pipeline, we first reduced the background noise of IMC by using a denoising method developed by us (IMCell^{XMBD}: A statistical approach for robust cell identification and quantification from imaging mass cytometry images. bioRxiv (2021)). However, after denoising, we still notice some “double-positive” cells, e.g., myeloid cells co-expressing CD3. Close examination of these cells in the original IMC image revealed that some of these cells are indeed myeloid cells locating next to T cells and the “double-positive” are results of imperfect cell segmentation (cell-A and cell-B, example figure shown below) which is very difficult to handle in our current processing pipeline. In our lab, we are currently developing a new computational method to address this “spatial doublet” problem.

2. Methods section on 'preparation and staining' is riddled with errors. Second line reads 65% instead of 65 degrees, line 6 states 70 degrees as room temperature.

Author's response: Thanks for the comments. We have corrected the errors in the revised manuscript.

3. It appears Figure 4D and 4E are mixed up in the figure legend

Author's response: Thanks for the comments. We have corrected it in the revised manuscript.

4. Figure 4D (in figure) doesn't have C2 (in Fig 4D).

Author's response: C2 is absent in Figure 4D because there is no significant pathway enriched in C2 samples based on the 0.05 threshold for adjusted P value. In the revised manuscript, we included this explanation in the caption of this figure as follows:

“Figure 4: (d) Gene set enrichment analysis (GSEA) of genes with up-regulated expression for each TME archetype. Significantly enriched gene sets (adjusted $P < 0.05$, Benjamini–Hochberg method) from MSigDB HALLMARK collection are shown. There is no significant pathway enriched in C2 samples based on the 0.05 threshold for adjusted P value.”

Reviewer #2 (Remarks to the Author):

Xiao et al reported an imaging mass cytometry analysis, coupled with bulk RNA-seq, to analyze the spatial pattern of pre-selected immune, stromal and tumor populations in a cohort of 26 melanoma patients from PUCH. The authors discovered different tumor microenvironment patterns that are related to response and non-response to anti-PD-1 therapy. In particular, they reported three separate "hot" TMEs that enrich different immune cell types: a CD8 related, high HLA-DR related (APC) and CD4 T and B cell related. In the cold TME, the enriched biological process is hypoxia (in either tumor or myeloid subset MC1/MC2). Finally, based on tumor matched RNA-seq data, the authors came up with a gene expression signature that were shown to be predictive of survival in multiple immunotherapy datasets of melanoma.

The reviewer is convinced that this work is of high interest to the field and highlights the importance of spatial information to predict response to anti-PD-1, even when only using a pre-selected markers of 35 proteins. My major concern is that most of the raw data and supporting processed values are nowhere to be found, which forces the reviewer to accept the results in the figures as is. The uploaded data to the figshare also requires log in, which may compromise the anonymity of the review process.

Major comments:

1. The distribution of the number of CT and IM ROIs per patient should be provided.

Author's response: Thanks for the suggestion. We have provided the exact locations of the selected ROIs in the original slides, and their distribution in the revised manuscript Supplementary Fig. S11 and Supplementary Table 2.

2. Both the overall and progression free survival data for each patient should be provided, along with other clinical parameters like clinical response, disease stage, previous treatment etc, in a supplementary table.

Author's response: Thanks for the suggestion. We have added this clinical data in Supplementary Table 2 of the revised manuscript.

3) The processed IMC data (cells after segmentation with raw marker intensity across 35 markers for all ROIs) should be provided to allow independent reproduction of the results listed here.

Author's response: Thanks for the suggestion and we apologize for the inconvenience in accessing the data in figshare. We have uploaded the raw image data and processed IMC data to Zenodo: <https://doi.org/10.5281/zenodo.6838169>

4) In the same vein, the reviewer cannot find the RNAseq data of this cohort being made available; the data should be included to ensure the reproducibility of the analysis. The github page only contain two R scripts. If the RNAseq is a part of the previous Cui et al publication (n=49 tumors), then the patient mapping from this set to the Cui et al set should be provided.

Author's response: Thanks for the suggestion. We have uploaded the *RNAseq* data to Zenodo: <https://doi.org/10.5281/zenodo.6838169>

5) The proximity to CD8 T cell analysis (Fig 4f) is not easy to decipher - there is no way for the reviewer to see, in the current data presentation, that MC1 and MC2 are closer to CD8 T cells in the immune cold TMEs. Also, the MC1 and MC2 was discussed to be avoiding the lymphocytes in the discussion in Fig 3.

Author's response: Thanks for this comment. In the revised manuscript, we added a new box plot to compare the minimum distance of different immune cell subtypes to CD8 T cells in different TME archetypes (Supplementary Fig. S8, as shown below). As can be observed in this figure, MC2 cells were closer to CD8 T cells in the immune cold TMEs compared to they were in immune hot TMEs. This result was summarized in the revised manuscript as follows:

“We observed significantly elevated accumulation of the HLA-DR⁻ subtype of myeloid cells MC2 in close contact with CD8⁺ T cells in all three immune “cold” TME archetypes (Fig. 4f, Supplementary Fig. S8), indicative of the potential role of these cells in creating an ICI resistant TME through suppressing effector T cells functionality.”

Supplementary Fig. S8. The nearest distance in μm between CD8⁺ T cells and other immune cells

As to the second question, it is indeed that HLA-DR⁻ myeloid cells showed avoidance to lymphocytes in non-responders. However, in consistence with the results from the proximity analysis above, myeloid

cells MC1 and MC2 did show interaction with CD8 T cells in non-responders (highlighted area 3 in Fig. 3b below). Note that compared to HLA-DR⁺ myeloid cells, significant interaction/avoidance between HLA-DR⁻ myeloid cells and lymphocytes appeared in much lesser number of ROIs (highlighted area 4 in Fig. 3b below). This is described in the revised manuscript as follows to reflect this observation:

“We also observed highly different interaction patterns of HLA-DR⁺ and HLA-DR⁻ myeloid cells with lymphocytes. Strong cognate interaction between the HLA-DR⁺CD11c⁺ myeloid cells (MC4) and lymphocytes can be observed in responders (Fig.3a, highlighted area 2) and, to a lesser extent, in nonresponders as well (Fig.3b, highlighted area 2; Supplementary Fig.S4a). In contrast, significant interaction/avoidance between HLA-DR⁻ myeloid cells and lymphocytes were observed in a much smaller number of ROIs (Fig. 3a-b, highlighted area 4). Interestingly, in nonresponders, HLA-DR⁻ myeloid cells showed avoidance to most lymphocytes except for MC1 and MC2, which showed interaction with CD8 T cells (Fig. 3b, highlighted area 3).”

Fig 3. Spatial analysis among cell phenotypes.

6) When the authors mentioned that the expression signatures of Bagaev et al or CIBERSORTx being uncorrelated with the IMC result, did the author tested basic correlation between the canonical T cell and myeloid specific transcript like CD8A/CD8B/CD14/CD3E etc between the bulk RNAseq and the IMC cell percentage?

Author’s response: Thanks for the suggestion. We used MCP-counter^[1] which uses average expression of canonical cell type markers for cell type abundance estimation from bulk RNA-seq data. Similar to the results from CIBERSORTx^[2], we found that although there exist correlations between cell abundance results from MCP-counter and the lymphocytes abundances from IMC, the highest correlation was not always from the target cell type of MCP-counter (Supplementary Fig. S10b). In addition, we didn’t find a good estimator for myeloid cell abundances from MCP-counter. We updated the revised paper as follows to reflect this observation:

“Similar observations can be made when we compared the cell type proportions estimated from IMC and those estimated by deconvolution methods from bulk RNA-seq data including MCP-counter^[1], which derives cell type abundance based on mean expression of canonical cell type markers, and CIBERSORTx^[2], of which cell type abundance is estimated using support vector regression on the basis of gene expression signatures of target cell types (Supplementary Fig.S10b-c).”

Supplementary S10b. Correlation between cell type abundance estimated by MCP-counter and IMC.

- [1] Becht, E. et al. Estimating the population abundance of tissue-infiltrating immune and stromal cell populations using gene expression. *Genome Biology* 17, 218 (2016).
- [2] Newman, A.M. et al. Determining cell type abundance and expression from bulk tissues with digital cytometry. *Nature Biotechnology* 37, 773–782 (2019).

7) *Is the H1 TME correlated with tertiary lymphoid structure?*

Author’s response: This is a good question. To explore this possibility, we asked our pathologist to exam all the ROIs and the corresponding H&E image, and 9 ROIs are determined as having TLS region based on its H&E and the protein (i.e., CD20, CD4, CD8, Ki67) expression pattern (Supplementary Fig S9). We have also included the TLS information in the heatmap from Fig.4a (as shown below). It can be seen from the result that of 9 ROIs containing TLS, 7 ROIs were determined as the H1 TME (0.78, 95% CI: 0.40-0.97), and 2 ROIs were from the H3 TME (0.22, 95% CI: 0.03-0.60). Also, the H1 TME contains 15 ROIs and 46.7% of these are determined as having TLS. In the revised manuscript, we have included this new result as follows:

“Recently, it was reported that tertiary lymphoid structure (TLS) is associated with improved clinical outcome [1,2,3,4]. To study if any of the above TME archetypes were associated with TLS, we asked a professional pathologist to identify TLS from the selected ROIs (Methods). In total 9 ROIs were determined as containing TLS (Supplementary Fig. S9). Among them, 7 ROIs are from H1 TME (0.78, 95% CI: 0.40-0.97), 2 ROIs are from the H3 TME (0.22, 95% CI: 0.03-0.60), while none of these TLS was from immune “cold” ROIs. In addition, 46.7% of ROIs from H1 TME contain TLS. These results

suggested that the immune “hot” TME archetypes identified in this study, in particular H1, were strongly associated with TLS.”

- [1] Schumacher, T. N. & Thommen, D. S. Tertiary lymphoid structures in cancer. *Science* 375, eabf9419 (2022)
- [2] Helmink, B. A. et al. B cells and tertiary lymphoid structures promote immunotherapy response. *Nature* 577, 549–555 (2020).
- [3] Petitprez, F. et al. B cells are associated with survival and immunotherapy response in sarcoma. *Nature* 577, 556–560 (2020).
- [4] Cabrita, R. et al. Tertiary lymphoid structures improve immunotherapy and survival in melanoma. *Nature* 577, 561–565 (2020).

Fig. 4a Heatmap showing scaled cell type abundance from the IM ROIs

8) Could MC4 monocyte derived DCs? the cluster's HLA-DR expression is significantly higher than other MC clusters. Independent staining of serial section using CD14, CD1C, CLEC10A and HLA-DR.

Author’s response: It is indeed a good suggestion. We tried to acquire the serial sections from samples enriched with MC4, and stain them with HLA-DR, CD14, CLEC10A, and CD1C. Unfortunately, we were not able to obtain good staining of CD1C and CLEC10A in our experience despite that we had tried different antibodies from different antibody vendors. Nevertheless, closer examination of clusters of HLA-DR⁺CD14⁺ cells appeared both on the IHC and IMC images (shown below) suggests that MC4 cells look more like a subtypes of macrophage highly expressing HLA-DR rather than monocyte derived DCs based on their morphology. Unfortunately, we were not able to arrive at a conclusion here due to missing important DC markers, and the actual cell subtype and role of this group of cells would deserve further investigation.

9) The prognostic significance of the 24-gene should be compared to IFN- γ /IMS ratio that was reported in Cui *et al.*

Author's response: Thanks for the suggestion. The paper from Cui *et al.* was targeting at finding a gene signature with good prediction performance of clinical outcomes of immunotherapy, and the signature was derived by comparing bulk RNA-seq of patients with different clinical outcomes in a combined discovery cohort derived from several immunotherapy studies. In this work, the signature was derived from patients with different TME archetypes based on the notion that the responses to immunotherapy are mainly driven by TME composition. We have compared the prediction/prognostic performances of both signatures and found their performances are very similar (shown in Table/Figure below). Unfortunately, at this moment we are still not able to tell which approach is superior in identifying potential responders from immunotherapy due to lack of data. More work needs to be done in future to answer this question.

	PUCH	Riaz17	Gide	Liu19
This paper	0.83 (0.67-0.96)	0.75 (0.54-0.91)	0.74 (0.59-0.89)	0.65 (0.49-0.80)
Cui et al.	0.81 (0.69-0.93)	0.76 (0.59-0.94)	0.83 (0.68-0.99)	0.66 (0.50-0.83)

From this paper:

From Cui et al. :

REVIEWERS' COMMENTS:

Reviewer #1 (Remarks to the Author):

The revised manuscript has improved the quality significantly. I am satisfied with this improved version. One comment is that although the authors have done a stat on the impact of the stage and site of disease, the numbers are small and I am not sure whether we could make a conclusion based on that data.

Reviewer #2 (Remarks to the Author):

The authors have made the necessary clarifications and revisions to the manuscript. They have also uploaded the raw and processed data to zenodo (is this a permanent data repository? it is important that the data remain accessible in the future).

Other than the above minor clarification, the reviewer has no other comment and recommends acceptance of the manuscript.

REVIEWERS' COMMENTS:

Reviewer #1 (Remarks to the Author):

The revised manuscript has improved the quality significantly. I am satisfied with this improved version. One comment is that although the authors have done a stat on the impact of the stage and site of disease, the numbers are small and I am not sure whether we could make a conclusion based on that data.

Author's response: Thanks for helping us to improve our manuscript. We agree with the reviewer and change the statement as follow: "In this cohort, melanoma subtypes were analyzed for association with immunotherapy outcome, and no factor shows a significant correlation with the clinical outcome, possibly due to our limited sample size (Supplementary Table 3)."

Reviewer #2 (Remarks to the Author):

The authors have made the necessary clarifications and revisions to the manuscript. They have also uploaded the raw and processed data to zenodo (is this a permanent data repository? it is important that the data remain accessible in the future).

Other than the above minor clarification, the reviewer has no other comment and recommends acceptance of the manuscript.

Author's response: Thanks again for the helpful comments, which helped us to improve our manuscript. The data repository is permanent for readers who are interested in.